# Clinical and Experimental Evidence for Patient Self-Inflicted Lung Injury (P-SILI) and Bedside Monitoring

**DOI:** 10.3390/jcm13144018

**Published:** 2024-07-10

**Authors:** Ines Marongiu, Douglas Slobod, Marco Leali, Elena Spinelli, Tommaso Mauri

**Affiliations:** 1Department of Anesthesia, Critical Care and Emergency, Fondazione IRCCS Ca’ Granda Ospedale Maggiore Policlinico, 20122 Milan, Italy; ines.marongiu@policlinico.mi.it (I.M.);; 2Department of Critical Care Medicine, McGill University, Montreal, QC H4A 3J1, Canada; 3Department of Pathophysiology and Transplantation, University of Milan, 20122 Milan, Italy

**Keywords:** lung injury, esophageal pressure, respiratory drive, monitoring

## Abstract

Patient self-inflicted lung injury (P-SILI) is a major challenge for the ICU physician: although spontaneous breathing is associated with physiological benefits, in patients with acute respiratory distress syndrome (ARDS), the risk of uncontrolled inspiratory effort leading to additional injury needs to be assessed to avoid delayed intubation and increased mortality. In the present review, we analyze the available clinical and experimental evidence supporting the existence of lung injury caused by uncontrolled high inspiratory effort, we discuss the pathophysiological mechanisms by which increased effort causes P-SILI, and, finally, we consider the measurements and interpretation of bedside physiological measures of increased drive that should alert the clinician. The data presented in this review could help to recognize injurious respiratory patterns that may trigger P-SILI and to prevent it.

## 1. Introduction: What Is P-SILI?

In patients admitted to the intensive care unit (ICU), spontaneous breathing (SB, i.e., the presence of patient inspiratory effort) is increasingly common before and during invasive mechanical ventilation, even when continuous sedation is administered. This is particularly relevant in patients with acute respiratory distress syndrome (ARDS), which accounts for ~10% of total admissions to intensive care and 23% of all mechanically ventilated patients [1]. According to the LUNG SAFE multicenter prospective cohort study, around 30% of patients with ARDS are spontaneously breathing already on the first day of admission to the ICU [1]. Fine tuning spontaneous breathing and the timing of intubation may be both crucial to the prognosis of these patients [2].

Regardless of whether tidal ventilation occurs completely passively or during SB, lung inflation requires the development of a distending transpulmonary (alveolar–pleural) pressure (P_L_) to overcome both the resistive and elastic forces that oppose lung expansion.

It has been known for decades that invasive mechanical ventilation can cause and perpetuate lung injury in a process referred to as ventilator induced lung injury (VILI) [3]. The primary mechanisms of VILI include excessive tidal volume (V_T_) and inspiratory P_L_ to inflate the lung, resulting in volutrauma and barotrauma, respectively, as well as cyclic opening and closing of collapsed lung units, referred to as atelectrauma. Importantly, these effects are more likely to occur in the setting of ARDS because of increased lung elastance and reduced volume of the aerated lung. The development of VILI results from a combination of the way clinicians set the ventilator and the severity of the underlying lung injury.

There are several important differences between completely passive and assisted ventilation, in which patients perform the work of breathing either totally or in part with the assistance of a ventilator. For example, spontaneous efforts play an important role in determining the regional distribution of tidal ventilation within the lung, favoring ventilation of dorsal lung zones [4]. This effect could maintain lung recruitment and may improve gas exchange depending on the magnitude of the effort and severity of the lung injury.

However, in patients with acute hypoxemic respiratory failure and, in particular, ARDS, a constellation of physiologic derangements [5], including pulmonary and systemic inflammation [6], contribute to increasing the central drive to breathe [7]. When neuromuscular function is preserved, increased respiratory drive increases the inspiratory effort. The resulting effort can become a predominant, and sometimes dramatic, contributor to an excessive increase in P_L_ during inspiration, even in the presence of acceptable oxygenation and low arterial CO_2_ [8].

Whether the lung is inflated through a combination of “positive” airway pressure provided by the ventilator and the development of “negative” pleural pressure due to a patient’s inspiratory efforts or only through the latter, the distending P_L_ increases. If respiratory drive increases dramatically, then so too can the increase in P_L_ during inspiration, thereby potentially generating the conditions for lung injury. This progression of lung injury and the subsequent deterioration in lung compliance and gas exchange (that may further increase respiratory drive) has been referred to as the “vicious cycle” of patient self-inflicted lung injury or P-SILI [9]. This “vicious cycle” refers to P-SILI auto-maintenance: high respiratory drive induces high effort determining high P_L_, which worsens lung injury. The consequences of worsening lung injury (inflammation, alterations in gas exchange, acidosis, and cyclic derecruitment) further trigger respiratory drive and the cycle re-starts (Figure 1).

Although the changes in P_L_ that develop during passive ventilation and SB are often compared [10], several differences exist between the mechanisms that lead to traditional VILI and P-SILI:During SB in the setting of a healthy lung, changes in pleural pressure brought about by diaphragm activation are homogenously distributed across the surface of the lung. This leads to an even distribution of regional P_L_ and lung inflation and has been referred to as the healthy lung exhibiting “fluid-like behavior” [11]. In contrast, the injured ARDS lung has been described as exhibiting “solid-like” behavior because intense diaphragm activation causes larger negative inspiratory pressure swings in the dorsal, collapsed lung region. As a result, significant regional variations in P_L_ induce excessive deformation of some lung regions and can cause a redistribution of ventilation within the lung during a single respiratory cycle.The redistribution of ventilation within the lung occurring at the early onset of strenuous inspiratory effort is a distinct mechanism that differentiates P-SILI from classical VILI and has been termed the “occult pendelluft” phenomenon. “Occult pendelluft” is the shift of gas from non-dependent to dependent regions during inspiration, in addition to the dorsal tidal volume coming from outside (ventilator or non-invasive support). In lung injured pigs, Yoshida et al. demonstrated that SB was associated with an early redistribution of ventilation from the non-dependent to the dependent lung occurring before the initiation of inspiratory flow from the ventilator [12]. What was particularly striking about this report is that the investigators also demonstrated that the dorsal V_T_ was nearly threefold higher during SB than during passive ventilation with neuromuscular blockade (NMB). The implication is that significant and potentially injurious levels of regional P_L_ may develop during SB, even at low global V_T_ and driving pressure.Another often-overlooked mechanism of injury specific to P-SILI is related to the hemodynamic changes that may result in pulmonary vascular injury. The fall in pleural pressure that occurs during inspiratory effort lowers right atrial pressure (referenced to atmosphere) and thereby decreases the downstream pressure that opposes venous return, favoring the return of blood to the right ventricle. At end-inflation, P_L_ is maximal and, particularly in the setting of reduced lung compliance, this can dramatically increase right ventricular afterload due to the increase in West non-zone 3 lung units [13,14]. This cyclic increase in RV preload followed by the increase in RV afterload may increase shear stress within the pulmonary vasculature and contribute to lung injury. This was the conclusion of an experimental study in which a detailed hemodynamic analysis was performed during a reproduction of the classic study on VILI by Webb and Tierney [15]. Although this study was performed under passive conditions, the cyclic exaggeration and interruption of RV filling and ejection during inspiration are expected to be even more prominent in the presence of decreased lung compliance and vigorous negative pleural pressure swings during SB [13,14,15,16,17].Finally, the inspiratory decrease in alveolar pressure to levels lower than PEEP increases the transmural pressure within the pulmonary vasculature, favoring fluid extravasation into the interstitial space. The tidal change in extravascular pressure [18] and exaggeration of pulmonary blood flow at high intravascular pressures [17] have both been shown to be potentially important contributors to lung edema that may be exaggerated during vigorous SB.

## 2. Does P-SILI Exist? Clinical Evidence

Studies of critically ill patients mostly provide indirect support for the existence and clinical importance of P-SILI (Table 1). Evidence of increased respiratory drive and effort, as well as their consequences (i.e., higher tidal volume or hypocapnia), have consistently been associated with adverse outcomes such as worsening respiratory failure and intubation, even after controlling for other clinical factors and disease severity. Moreover, a randomized controlled trial of an intervention that removed respiratory effort demonstrated benefit [19].

In the setting of more severe forms of ARDS, maintenance of SB with non-invasive ventilation (NIV) has been associated with worse outcomes. Prospective observational data from the LUNG SAFE study demonstrated that among patients with ARDS and a PaO_2_/FiO_2_ < 150 who were treated initially with at least 2 days of NIV, over 40% of patients required intubation [20]. Hospital mortality in patients who failed initial NIV therapy was very high (45%). In a propensity score matched sample, ICU mortality was higher in patients with moderate-severe ARDS who were treated initially with NIV compared to invasive mechanical ventilation.

Among patients with acute hypoxemic respiratory failure who are treated initially with NIV, physiologic parameters that indicate increased respiratory drive and effort have been associated with failure of NIV, indicating the possibility of a causal relationship between high P_L_ and worsening lung injury. In a study of 62 patients with acute hypoxemic respiratory failure, primarily due to pneumonia, and 75% of whom had a diagnosis of ARDS who were initially treated with NIV, greater V_T_ was independently associated with need for intubation, after controlling for severity of illness and severity of hypoxemia [21].

Tonelli et al. demonstrated that among patients with acute hypoxemic respiratory failure, a lack of decrease in respiratory effort, defined as a reduction in the inspiratory esophageal pressure swing <10 cmH_2_O, within 2 h after application of NIV was highly predictive of intubation [22].

Similarly, in a recent retrospective study of over 1000 patients with hypoxemic respiratory failure receiving initial support with non-invasive ventilation, a non-linear relationship between the initial PaCO_2_ prior to NIV initiation and need for intubation was identified, with a sharp increase in risk of need for intubation once PaCO_2_ values decreased below 32 mmHg [23]. Importantly, this relationship remained 1–2 h after application of NIV, confirming that patients who continue to demonstrate increased respiratory drive and effort during NIV are at risk of worsening.

Among patients with COVID-19 pneumonia who were initially treated with continuous positive airway pressure or NIV, Coppola et al. estimated total lung stress as: (ΔPaw − ΔPes) + (PEEP ∗ 0.7)
with ΔPaw being the applied pressure support and ΔPes being the inspiratory change in esophageal pressure [24]. The factor of 0.7 represents an estimate of the proportion of the applied positive end-expiratory airway pressure (PEEP) transmitted to the lung. They found that total lung stress was significantly higher among patients who required intubation. Moreover, trends of daily measurements of total lung stress demonstrated that patients who never required intubation manifested a decrease in total lung stress whereas patients who failed NIV demonstrated no change or increasing values over time.

Additional clinical evidence supporting the existence of P-SILI comes from studies of spontaneously breathing intubated patients. A recently published retrospective study of a general population of mechanically ventilated patients [25] demonstrated that greater values of the 100 ms airway occlusion pressure (P_0.1_), a simple bedside measurement of increased respiratory drive, were significantly associated with greater degrees of dyspnea and higher 90-day mortality. Importantly, the association remained significant in multivariable modeling that controlled for several clinical factors, including severity of illness, blood gas values, and respiratory rate, suggesting that increased respiratory drive plays a unique role in the worsening of lung injury.

In a study of 340 patients performed in 2010, Papazian et al. demonstrated that, compared to a placebo, early administration of a 48 h infusion of a neuromuscular blocker (NMB) in patients with moderate-severe ARDS reduced mortality and increased ventilator-free days [19]. Although respiratory effort was not directly assessed in this study, the proposed mechanism of benefit in the intervention group was the removal of patient respiratory efforts, known to be frequently elevated in patients with ARDS despite not always being clinically evident [26]. This study also demonstrated earlier and significantly more frequent development of pneumothorax and barotrauma in the placebo group, despite there being no substantial difference between V_T_, plateau pressure, respiratory rate, and minute ventilation. This suggests that ventilator asynchrony, the distribution of P_L_, and occult pendelluft may have been important underlying mechanisms of injury that were eliminated in the NMBA group. However, in 2019, a re-evaluation of systemic early neuromuscular blockade investigators re-evaluated the impact of NMB in patients with moderate-severe ARDS [27] and found no difference in mortality. However, there was a trend towards less barotrauma in the NMB group. Compared to the 2010 study, this study used a strategy of lighter sedation in the control group and higher set PEEP. The apparent differences between the two studies might be explained by the fact that application of higher PEEP has been associated with lower respiratory drive and effort in patients with hypoxemic respiratory failure [5,28], whereas sedation depth is poorly predictive of respiratory drive [5,29].

**Table 1 jcm-13-04018-t001:** Review of clinical evidence supporting the presence of P-SILI.

Clinical Studies	Clinical Setting	Type of Ventilatory Support	Sample Size	Main Results
Papazian: *N. Engl. J. Med.* **2010**, *363*, 1107–1116. [19]	Acute respiratory distress syndrome.	Invasive mechanical ventilation.	340	Administration of neuromuscular blockade decreased the occurrence of barotrauma and increased adjusted 90-day survival and number of ventilator-free days.
Carteaux: *Crit. Care Med.* **2016**, *44*, 282–290. [21]	Acute hypoxemic respiratory failure.	Non-invasive ventilation.	62	Expired tidal volume independently associated with failure of non-invasive ventilation.
Bellani: *Am. J. Respir. Crit. Care Med.* **2017**, *195*, 67–77. [20]	Acute respiratory distress syndrome.	Non-invasive ventilation.	436	Failure of non-invasive ventilation occurred in 47.1% of patients with severe ARDS and NIV use was independently associated with increased ICU mortality.
Tonelli: *Am. J. Respir. Crit. Care Med.* **2020**, *202*, 558–567. [22]	Acute hypoxemic respiratory failure.	Non-invasive ventilation.	30	Reduction in the esophageal pressure swing by 10 cm H2O or more after 2 h of non-invasive ventilation strongly associated with avoidance of intubation.
Coppola: *Intensive Care Medicine* **2021**, *47*, 1130–1139. [24]	COVID-19 pneumonia.	Continuous positive airway pressure or non-invasive ventilation	140	Total lung stress independently associated with failure of non-invasive respiratory support.
Xu: *BMC Pulm. Med* **2024**, *24*, 228. [23]	Acute hypoxemic respiratory failure.	Non-invasive ventilation.	1029	Lower PaCO_2_ non-linearly associated with increased intubation risk.
Le Marec: *J. Respir. Crit. Care Med.* **2024** [25].	Patients receiving mechanical ventilation in the intensive care unit for more than 24 h.	Invasive mechanical ventilation.	260	Elevated P_0.1_ independently associated with increased mortality.

## 3. Does P-SILI Exist? Experimental Evidence

Considerable experimental evidence has helped to identify the mechanisms underlying P-SILI. In the last decades, the role of SB has been investigated in different animal models of lung injury and several harmful effects of SB have been elucidated (Table 2).

Compared to controlled mechanical ventilation, which requires sedation and muscle paralysis with cranial displacement of the diaphragm and possible development of lung collapse in dependent lung regions, SB improves dorsal lung aeration. On the other hand, SB appears to be associated with injurious mechanisms, some of which may be like those classically described for VILI. Indeed, inflammatory mediator release (such as TNFα, IL-6, and prostacyclin) has been documented in isolated mouse lungs subjected to hyperventilation, regardless of being ventilated by positive or negative pressure [30]. During ARDS, negative pleural pressure deflections generated by strong respiratory efforts may recruit atelectatic “unstable” lung regions, with the risk of repetitive opening and closing of these regions (atelectrauma). This process may promote P-SILI and the vicious cycle of high respiratory drive and effort, which perpetuates lung injury (biotrauma) and further increases respiratory drive.

In a recent study conducted on a mice model of LPS induced ARDS + resistive spontaneous breathing (inducing higher respiratory effort, higher PL, and lower alveolar pressure), the resistive breathing caused a progression in lung injury in ARDS mice, when compared to resistive breathing in healthy animals and ARDS without resistive breathing. Mice with LPS induced ARDS + resistive spontaneous breathing showed distinctive pulmonary pathological changes (i.e., congestion and edema) and increased IL-1β, IL-6, TNF-α, and total protein levels in bronchoalveolar lavage [31].

A pivotal experimental study showing the injurious consequences of high respiratory drive during SB was conducted by Mascheroni et al. in the 1980s [32]. They infused sodium salicylate into the cisterna magna of 26 adult sheep to induce spontaneous hyperventilation. Among them, 16 were left spontaneously breathing room air, while 10 were sedated and paralyzed under physiologic controlled mechanical ventilation. Five animals received a placebo and were used as controls. After 24 h, gas exchange, respiratory mechanics and gross lung injury appearance (post-mortem) were compared between the groups. The sheep receiving sodium salicylate and induced to spontaneously hyperventilate developed major alterations (i.e., at gross inspection, development of atelectasis in up to 30% of the lungs, not recruitable with insufflation to 35 cmH_2_O) while the other animals remained normal. These findings demonstrate that hyperventilation is a key factor in the development of lung injury, even in healthy lungs.

Many years later, with a renewed interest in the field of SB and progress in the field of experimental and clinical research, Yoshida et al. conducted a study on a rabbit model of lung injury induced by lung lavage with normal saline until reaching a constant PaO_2_/FiO_2_ of 100 mmHg [33]. The aim was to demonstrate that SB may generate high P_L_ during assisted mechanical ventilation in damaged lungs. They found that a combination of relatively high tidal volume (8–10 mL/kg) during assisted pressure-controlled ventilation and strong SB effort caused more severe histologic lung injury in comparison to moderate tidal volume + low respiratory effort and to low tidal volume + strong or low respiratory effort, even when the plateau pressure was limited to <30 cmH_2_O.

The same group subsequently compared SB to controlled mechanical ventilation in mild and severe lung injury models. They found that SB was beneficial in the mild injury group in terms of oxygenation, respiratory mechanics, and lung aeration. In the severe injury + SB group, they observed higher P_L_ and a greater amount of cyclic collapse at end-expiration in the dependent regions, resulting in no improvement in oxygenation and worsening histological injury [4].

Yoshida et al. went further to document a distinct trigger of lung injury which is typical of SB: occult pendelluft [12]. In injured pig lungs, SB generated negative pleural pressure, causing a shift of alveolar gas from non-dependent to dependent lung regions and thus an overstretch of dependent regions, but no changes in global tidal volume. This is due to the “solid-like” behavior of the injured lungs, which implies an inhomogeneous distribution of pleural pressure and distending forces on the surface of the lung; therefore, larger negative pleural pressure deflections caused by contraction of the diaphragm are inadequately transmitted to the lung, leading to higher local lung stress in the dependent regions. Along the same lines, in a subsequent study conducted on rabbits and pigs with ARDS, Morais et al. demonstrated that strong spontaneous efforts (vs. muscle paralysis) predominantly injured the dependent lung. Moreover, at low PEEP there was a wide gradient of negative pleural pressure from non-dependent to dependent units, with tidal recruitment in the dependent lung (the stretch in the dependent lung regions corresponded to that applied by a VT of 14 mL/kg during muscle paralysis) and higher histological injury in those regions. Application of higher PEEP decreased effort also by electromechanical uncoupling of the diaphragm, leading to a reduction in the vertical gradient of pleural pressure from non-dependent to dependent regions (i.e., lower local lung stretch in the dependent regions), and decreased inflammation across all lung regions [28].

The heterogeneous distribution of ventilation during SB in the early phase of ARDS was described in a study by Bachmann et al. in a model of partial surfactant depletion and lung collapse in pigs [34]. The study compared the effects of short vs. prolonged duration of SB. The prolonged SB group showed large esophageal pressure swings, a predominantly dorsal distribution of ventilation, and an inhomogeneous temporal and spatial distribution of ventilation observed with EIT imaging, all factors involved in the progression of lung injury. Ventilation distribution became homogenous after a switch to protective controlled mechanical ventilation, but lung histological damage was not reverted. Their results suggest that prolonged strong inspiratory effort could worsen lung injury, while controlled mechanical ventilation could be beneficial, when applied early. The same researchers demonstrated that, in an animal model of severe ARDS supported with extracorporeal membrane oxygenation (ECMO) to control respiratory drive, SB, characterized by lower PL swings, high respiratory rates, high PEEP, and very low tidal volumes, resulted in an increase in dorsal regional ventilation without evidence of pendelluft or worsened lung injury compared to controlled ventilation [35]. These findings suggest that maintenance of “safe” SB could be crucial for the outcome of ARDS.

Overall, experimental experience supports the notion that stronger spontaneous efforts are associated with more severe P-SILI. Additional indirect evidence of the central role of effort is that strategies that reduce effort during SB (like higher PEEP [28] or ECMO [35]) limit progression of P-SILI.

## 4. Bedside Monitoring to Prevent P-SILI

To prevent P-SILI, it is crucial to monitor the respiratory drive and effort at the bedside. It could also be relevant to recognize respiratory patterns which lead to lung damage (i.e., inhomogeneous ventilation, high tidal volume, respiratory asynchrony, and so on) [7].

The respiratory drive is the neural input, i.e., the amplitude and frequency of the signal generated from the brain, while the effort is the activation of the respiratory muscles induced by that signal. Input and response do not always correspond in terms of intensity; in the presence of altered neuromuscular transmission or weakness of the respiratory muscles, for example, high respiratory drive might not generate a strong contraction of the muscles, and the variations in tidal volume and lung pressures would not reflect the neural drive [36].

In non-intubated patients, the clinical signs that suggest high respiratory drive and effort are dyspnea, recruitment of accessory inspiratory and expiratory muscles, and tachypnea. However, although these findings suggest high respiratory drive and effort, they cannot quantify their magnitude.

### 4.1. How to Quantify the Respiratory Drive

#### 4.1.1. Diaphragm Electrical Activity (EAdi)

The EAdi signal is recorded by a suitable nasogastric catheter with multiple electrodes placed at the level of the diaphragm and connected to a dedicated ventilator software. The EAdi reveals crural diaphragm activity through the measure of the electrical field produced by motor neurons [37]. To assess the intensity of respiratory drive, the inspiratory EAdi peak value (EAdi_PEAK_), the slope of the EAdi from the beginning of inspiration to peak (EAdi_SLOPE_), and the neural inspiratory time (Ti_NEUR_) can be measured [38]. Unfortunately, reference normal values of EAdi_PEAK_ are unknown because of the large interindividual variability of the parameter. EAdi is mostly useful to assess variations in respiratory drive over time, and is not as helpful for identifying increased respiratory drive [7]. The ratio between the change in airway pressure during a brief end-expiratory occlusion and EAdi_PEAK_ or between tidal volume and EAdi_PEAK_ are absolute indexes of the coupling between neural respiratory drive and, respectively, mechanical (neuro-muscular efficiency, NME) and ventilatory responses (neuro-ventilatory efficiency, NVE) [39,40]. The muscular inspiratory pressure can then be calculated as the product of EAdi_PEAK_ ∗ NME [41] and the muscular pressure time product as the area under this signal. A limitation is that EAdi estimates only the neural drive to the diaphragm and cannot detect the activation of accessory muscles, which may be relevant when breathing effort is elevated.

#### 4.1.2. P0.1

In intubated patients, the airway occlusion pressure at 100 ms (P0.1) provides a more comprehensive evaluation of the respiratory drive. P0.1 is the negative pressure generated by all the inspiratory muscles during the first 0.1 s of inspiration against an occluded airway (infinite elastance and resistance). P0.1 is not affected by muscle weakness or flow resistance [36,42]. Normal values lie between 0.5 and 1.5 cmH_2_O, and the upper threshold is 3.5 cmH_2_O, as values above this threshold correlate well with high drive (EAdi) and effort (pressure-time product >200 cmH_2_O∙s/min) [43]. Recently, in a multicenter cohort study that evaluated dyspnea in communicative mechanically ventilated patients recovering from acute respiratory failure, Le Marec et al. found an association between higher P0.1 and dyspnea; moreover, higher P0.1 levels were independently associated with mortality and duration of mechanical ventilation [25].

The methods to monitor the respiratory drive are summarized in Table 3.

### 4.2. How to Quantify Respiratory Effort

#### 4.2.1. ΔPes and P_mus_

During SB, contraction of the inspiratory muscles produces a deflection in esophageal pressure, which corresponds to the change in Ppl and reflects the magnitude of effort [44]. Moreover, the pressure generated by the respiratory muscles (P_mus_) can be calculated as the difference between the static recoil pressure of the chest wall and ΔPes. Physiological values of ΔPes during SB stay between 5 and 10 cmH_2_O.

#### 4.2.2. WOB and PTP

More sophisticated indices of breathing effort, correlated to the energy expenditure of the respiratory muscles, are the work of breathing (WOB), which corresponds to the integral of the product of Pmus and the tidal volume, and the pressure time product (PTP), which corresponds to the integral of the product of Pmus and time [44,45].

#### 4.2.3. ΔP_occ_

An easy to obtain index of effort in intubated patients is the ΔP_occ_, which is the negative airway pressure swing exerted by the patient during an end-expiratory occlusion. In patients with respiratory failure, two studies have demonstrated a correlation between ΔP_occ_ and Pmus (with average values equal to 0.75 ∗ ΔP_occ_) and ΔPes (with average values equal to 0.66 ∗ ΔP_occ_) [46,47]. 

#### 4.2.4. Tidal Swing of CVP

Since breathing is accompanied by cyclic variations in intrathoracic pressure, all the structures inside the thorax are subjected to pressure oscillations during ventilation. In the absence of an esophageal balloon catheter, the tidal swing of central venous pressure could be a surrogate for ΔPes and an adjunctive tool to estimate patient effort during SB [48]. Protti et al. suggested a cut off of 8 mmHg for ΔCVP to predict ΔPes > 10 cmH_2_O [49]. In patients with a pulmonary artery catheter, tidal swings in the pulmonary artery occlusion pressure provide an estimate of the change in ΔPes, which may be more reliable than ΔCVP [50].

#### 4.2.5. Diaphragm Ultrasound

The thickening of the diaphragm (i.e., thickening fraction, TF) during inspiration has shown a correlation with indexes of effort [51]. Suggested cut off values of TF for high effort are around 30%. In a recent study, conducted on a cohort of patients with respiratory failure in pressure support ventilation during their weaning phase, the authors compared the implementation of the two indices (diaphragm TF vs. ΔCVP) to evaluate inspiratory effort, as detected by the ΔPes. Both ΔCVP and the diaphragm TR results correlated with the inspiratory effort, but the ΔCVP had a stronger correlation [52].

The methods to monitor the respiratory effort are summarized in Table 4.

### 4.3. How to Monitor Dangerous Breathing Patterns

#### 4.3.1. Tidal Volume and Respiratory Rate

Increased respiratory drive, when ventilatory demand is increased and muscle strength is preserved, is firstly reflected by an initial increase in tidal volume with unchanged inspiratory time (Ti) and higher mean inspiratory flow (Vt/Ti) [53]. When the respiratory drive further increases or the patient develops muscular weakness, the respiratory rate increases, with reduced expiratory time. Another useful index, initially described as a weaning predictor, is the rapid shallow breathing index (RSBI) [53,54], that is, the ratio of respiratory rate (RR) to tidal volume (VT). A threshold value of >105 breaths/min/L might indicate unsatisfied ventilatory demand.

#### 4.3.2. Asynchronies

Specific patient-ventilator asynchrony patterns are correlated with high (or low) respiratory drive and effort. For example, missed effort and autotriggering are usually indicative of insufficient respiratory drive. Premature cycling and double triggering usually are epiphenomena of high respiratory drive and effort [55]. Recently, reverse triggering has emerged as a novel asynchrony during which drive is reversely activated by the mechanical breath and is associated with excessive tidal volume potentially leading to P-SILI [56].

#### 4.3.3. Distribution of Ventilation with EIT

High respiratory drive and effort may result in an inhomogeneous distribution of ventilation measured by electrical impedance tomography (EIT), with “occult pendelluft”, tidal recruitment of the lung, and excessive stress in the dependent regions. EIT can be integrated with other available monitoring, i.e., esophageal pressure swings, to provide comprehensive bedside monitoring during SB [57] (Figure 2). Other offline analysis of EIT data include the calculation of a series of parameters, such as the global inhomogeneity index (GI), an index of lung inhomogeneity, or the regional ventilation delay (RVD), which represents the temporal delay of regional ventilation and is associated with tidal recruitment [34]. Each of these parameters can be accentuated by strenuous inspiratory efforts.

The estimation of respiratory drive and breathing effort can be difficult, especially in non-intubated patients lacking invasive monitoring. In this scenario, it may help to potentially recognize clinical risk factors related to high respiratory drive and effort. In a clinical study conducted by Spinelli et al., the authors investigated the relation between clinical risk factors associated with increased respiratory drive and P0.1 in a population of intubated patients with acute respiratory failure or ARDS on pressure support ventilation. The independent factors predicting higher respiratory drive were diagnosis of ARDS, lower PaO_2_/FiO_2_ and higher ventilatory ratio (both indices of altered ventilation/perfusion matching), lower arterial pH, and lower set PEEP [5].

In a very recent study, Protti et al. developed two very simple models for estimating high respiratory effort from simple bedside clinical data. They studied a cohort of patients with high-flow oxygen therapy, and they identified base excess, respiratory rate, and PaO_2_:FiO_2_ as the clinical variables correlated with higher effort (ΔPes > 10) [60].

## 5. Conclusions

Clinical and experimental evidence support the construct that strong inspiratory efforts in the setting of lung injury may worsen the injury. While the clinical research provides indirect evidence of P-SILI, the experimental setting helps to elucidate some of the mechanisms involved in its development. Some of the determinants of P-SILI are similar to VILI, i.e., high P_L_ and tidal volume with unpredictable stress and strain; while others are unique to P-SILI, i.e., occult pendelluft and cyclic opening and closing of unstable dorsal regions. During SB in ARDS, dedicated monitoring at the bedside may help with earlier recognition of the injurious triggers of P-SILI and to discern when maintenance of SB may be unsafe.

## Figures and Tables

**Figure 1 jcm-13-04018-f001:**
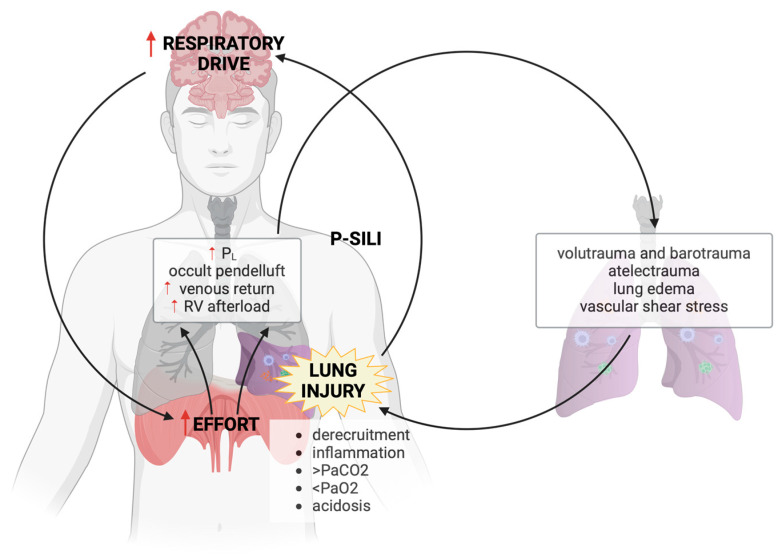
The “vicious cycle” of P-SILI. Lung injury triggers alterations in gas exchange, derecruitment, and activation of inflammation, all contributing to an increase in the respiratory drive. High respiratory drive produces high effort which in turns causes large deflections in pleural pressure with lung stress and hemodynamic changes, resulting in volu- and barotrauma, atelectrauma, lung edema and vascular shear stress. All these mechanisms further increase lung injury, and the cycle re-starts. P_L_, transpulmonary pressure; RV, right ventricle; P-SILI, patient self-inflicted lung injury. [Created with BioRender.com, accessed on 30 June 2024].

**Figure 2 jcm-13-04018-f002:**
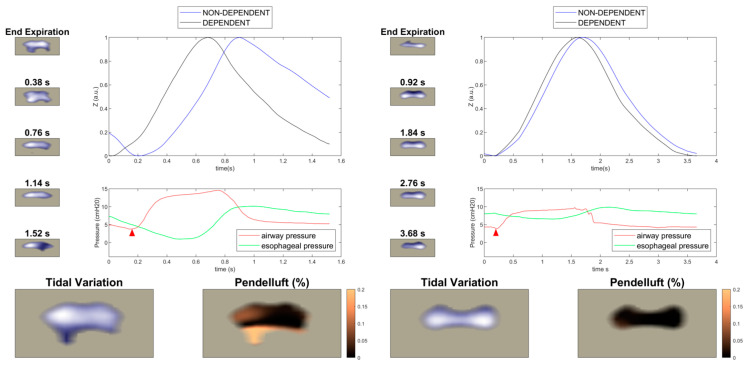
Integrated monitoring with EIT and esophageal pressure during pressure support ventilation. Impedance and pressure traces of two patients undergoing pressure support ventilation at PEEP 4 cmH_2_O. A representative respiratory cycle for each patient has been sampled and imaged starting from end-expiration (see image columns at the left of *y*-axes). In the patient on the left side, it is evident that the dependent lung regions (black trace) start inflation earlier than the non-dependent lung regions (blue trace), preceding the trigger at the ventilator (red arrowhead). Impedance traces have been rescaled to fit the 0 ÷ 1 interval as in [58], for clarity. This results in increased pendelluft (see images in red colorscale), as quantified by the method here referenced [59], and corresponds to a dynamic swing in esophageal pressure of about 6 cmH_2_O (green trace) at a nominal pressure support level of 10 cmH_2_O (red trace). In the patient on the right side, the time lag is not as evident and pendelluft is markedly lower. Accordingly, the dynamic swing in esophageal pressure is roughly 2 cmH_2_O at a pressure support level of 6 cmH_2_O.

**Table 2 jcm-13-04018-t002:** Review of experimental evidence supporting the presence of P-SILI.

Experimental Studies	Clinical Setting	Type of Ventilatory Support	Sample Size	Main Results
von Bethmann: *Am. J. Respir. Crit. Care Med*. **1998**, *157*, 263–272. [30]	Isolated hyperventilated and perfused mouse lung.	Positive pressure ventilation (PPV) or negative pressure ventilation (NPV).	12	Hyperventilation resulted in an increased expression of TNFα and IL-6 mRNA, and prostacyclin release into the perfusate.
Cai: *Biochem. Biophys. Rep*. **2024**, *38*, 101726. [31]	LPS induced ARDS + tracheal banding in female mice.	Resistive spontaneous breathing (RSB).	60	RSB exacerbated lung injury in ARDS: more congestion and edema, more severe inflammatory cell infiltration, and increased IL-1β, IL-6, TNF-α, and total protein levels in BALF.
Mascheroni: *Intensive Care Med.* **1988**, *15*, 8–14. [32]	Hyperventilation induced by sodium salicylate infusion in the cisterna magna of adult sheep.	Spontaneous breathing.	31	Hyperventilation by SB induced alterations in gas exchange, a decrease in the static compliance of the respiratory system, and atelectasis.
Yoshida: *Crit Care Med*. **2012**, *40*, 1578–1585. [33]	Acute lung injury induced by lung lavage with 25 mL/kg of normal saline in rabbits.	Invasive mechanical ventilation + spontaneous breathing.	32	High P_L_ generated by strong spontaneous breathing effort worsened lung injury.
Yoshida: *Crit Care Med* **2013**, *41*, 536–545. [4]	Mild lung injury induced by lung lavage and severe lung injury induced by lung lavage + injurious mechanical ventilation in rabbits.	Invasive mechanical ventilation + spontaneous breathing.	28	SB worsened lung injury in the severe lung injury group, while muscle paralysis was protective.
Yoshida: *Am. J. Respir. Crit. Care Med.* **2013**, *188*, 1420–1427. [12]	Acute lung injury in pigs.	Invasive mechanical ventilation + spontaneous breathing.	7	Spontaneous breathing effort during mechanical ventilation caused pendelluft and overstretch during early inflation, with more negative local Ppl in dependent regions.
Morais: *Am. J. Respir. Crit. Care Med.* **2018**, *197*, 1285–1296. [28]	Lung injury induced by lung lavage + injurious mechanical ventilation in rabbits.	Invasive mechanical ventilation + spontaneous breathing.	28	Strong spontaneous effort at low PEEP injured the dependent lung, while high PEEP was protective.
Bachmann: *Sci. Rep*. **2022**, *12*, 12648. [34]	Acute lung injury induced by lung lavage with 30 mL/kg of isotonic saline in pigs.	Pressure support ventilation or controlled mechanical ventilation.	18	Prolonged SB caused progression of lung injury, while early muscle paralysis and controlled mechanical ventilation could be beneficial.
Dubo: *Anesthesiology* **2020**, *133*. [35]	Lung injury induced by lung lavage + injurious mechanical ventilation in pigs.	Invasive mechanical ventilation + spontaneous breathing during ECMO.	12	SB during ECMO in severe ARDS did not result in worsened lung injury if compared to controlled mechanical ventilation.

**Table 3 jcm-13-04018-t003:** Monitoring of the respiratory drive.

Monitoring Method	Main Measures	Physiological Range	Advantages	Limitations
Neural activity of the diaphragm	EAdi_PEAK_	Lack of absolute values. In SB healthy subjects 13–21 μV [38].	Close to the neural drive, useful to assess change of the neural drive over time, EAdi does not require intubation.	Interindividual variability (no reference values), cannot detect activation of respiratory muscles apart from diaphragm.
	Ti_NEUR_	In SB healthy subjects 1.5–2 ms [38].		
	NVE = EAdi_PEAK_/Vt	Lack of absolute values [39].		
	NME = EAdi_PEAK_/ΔPaw	Lack of absolute values [40].		
	EAdi_PEAK_ ∗ NME	Lack of absolute values [40].		
Airway occlusion pressure	P0.1	1.0–3.5 cmH_2_O [43]	Not affected by muscle weakness or flow resistance	Requires intubation, breath-to-breath variability, can change according to the ventilator mode

**Table 4 jcm-13-04018-t004:** Monitoring of the respiratory effort.

Monitoring Method	Main Measures	Physiological Range	Advantages	Limitations
Esophageal pressure swings	ΔPes	5–8 cmH_2_O [44]	Good indicator of effort, easy to obtain at the bedside and in non-intubated patients	Cannot discriminate the effort required to expand the chest wall.
	P_mus_	5–10 cmH_2_O [44]	Best indicators of effort	Requires intubation and measurement of elastic chest wall recoil pressure under passive conditions.
	WOB	0.35–2.4 j min^−1^ [44]		
	PTP	80–200 cmH_2_O s min^−1^ [43]		
Negative airway pressure swing during end-expiratory occlusion	ΔP_occ_	6–13 cmH_2_O [46]	Good correlation with ΔPes and P_mus_, easy to obtain in intubated patients.	Requires intubation and collaboration of the patient.
Tidal swing of CVP	ΔCVP	0–8 mmHg [49]	Good correlation with ΔPes, useful in the absence of an esophageal balloon catheter	Depends on volemic state of the patient.
Diaphragm ultrasound	Thickening fraction	15–30% [51]	Easy to obtain at the bedside and in non-intubated patients, cheap.	Does not account for inspiratory and expiratory muscle activation.

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
