# Peer review of "Clinical and Experimental Evidence for Patient Self-Inflicted Lung Injury (P-SILI) and Bedside Monitoring"

_jcm, 2024, doi:10.3390/jcm13144018_

Round 1

Reviewer 1 Report

Comments and Suggestions for Authors

The mechanisms that lead to ventilation-induced lung injury (VILI), essentially attributable to an excess of Vt and trans-pulmonary pressure, have been known for some time. Those that correlate damage from ventilation to spontaneous breathing and its determinants, drive and respiratory effort are the result of observations and elaborations over the last ten years. The Covid experience was an important opportunity to delve deeper into the topic. The importance of these acquisitions cannot be underestimated. Think for example of the positioning of non-invasive respiratory support in the treatment of acute "de novo" respiratory failure: the available guidelines do not express recommendations in this regard considering the high risk of early death observed in many studies in patients who failed the method compared to those who were immediately treated with conventional ventilation. Today we know that a large part of those failures can be explained by the vicious cycle of lung damage induced by P-SILI in the spontaneously breathing patient. Therefore, the possibility of monitoring these patients at bedside level with instruments that allow a reliable quantification of the drive and respiratory effort, treating non-invasively only patients who remain within precise values ​​of the same or in whom it is possible to maintain these ranges through a adequate sedation represents in some way a paradigm shift. This review represents an elegant consultation tool for those who want to approach the experimental and clinical evidence of P-SILI and have an overview of what the monitoring methods of its determinants could be. For this reason it will be much appreciated by the Journal's readers. As the only minimal suggestion for improvement, I suggest introducing two tables (one for respiratory drive and one for respiratory effort) with the characteristics of the main monitoring methods, the reference values, the advantages and disadvantages.

Author Response

  1. The mechanisms that lead to ventilation-induced lung injury (VILI), essentially attributable to an excess of Vt and trans-pulmonary pressure, have been known for some time. Those that correlate damage from ventilation to spontaneous breathing and its determinants, drive and respiratory effort are the result of observations and elaborations over the last ten years. The Covid experience was an important opportunity to delve deeper into the topic. The importance of these acquisitions cannot be underestimated. Think for example of the positioning of non-invasive respiratory support in the treatment of acute "de novo" respiratory failure: the available guidelines do not express recommendations in this regard considering the high risk of early death observed in many studies in patients who failed the method compared to those who were immediately treated with conventional ventilation. Today we know that a large part of those failures can be explained by the vicious cycle of lung damage induced by P-SILI in the spontaneously breathing patient. Therefore, the possibility of monitoring these patients at bedside level with instruments that allow a reliable quantification of the drive and respiratory effort, treating non-invasively only patients who remain within precise values ​​of the same or in whom it is possible to maintain these ranges through a adequate sedation represents in some way a paradigm shift. This review represents an elegant consultation tool for those who want to approach the experimental and clinical evidence of P-SILI and have an overview of what the monitoring methods of its determinants could be. For this reason it will be much appreciated by the Journal's readers. As the only minimal suggestion for improvement, I suggest introducing two tables (one for respiratory drive and one for respiratory effort) with the characteristics of the main monitoring methods, the reference values, the advantages and disadvantages.

We thank the revisor for the comments and we really hope our review will help the clinicians in the management of patients with acute respiratory failure during SB. We agree with the reviewer that data on monitoring need to be summarized to be easily available to the reader. For this reason, we added table 3 and table 4 about monitoring methods.

Table 3. Monitoring of the respiratory drive.

Monitoring method

Main measures

Physiological range

Advantages

Limitations

Neural acitivity of the diaphgram

EAdiPEAK

Lack of absolute values. In SB healthy subjects 13-21 mV [38].

Close to the neural drive, useful to assess change of the neural drive over time, EAdi does not require intubation.

Interindividual variability (no reference values), cannot detect activation of respiratory muscles apart from diaphragm.

TiNEUR

In SB healthy subjects 1.5-2 ms [38].

NVE = EAdiPEAK/ Vt

Lack of absolute values [39].

NME = EAdiPEAK/ΔPaw

Lack of absolute values [40].

EAdiPEAK * NME

Lack of absolute values [40].

Airway occlusion pressure

P0.1

1.0-3.5 cmH2O [43]

Not affected by muscle weakness or flow resistance

Requires intubation, breath-to-breath variability, can change according to the ventilator mode

Table 4. Monitoring of the respiratory effort.

Monitoring method

Main measures

Physiological range

Advantages

Limitations

Oesophageal pressure swings

ΔPes

5-8 cmH2O [44]

Good indicator of effort, easy to obtain at the bedside and in non-intubated patients

Cannot discriminate the effort required to expand the chest wall.

Pmus

5-10 cmH2O [44]

Best indicators of effort

Requires intubation and measurement of elastic chest wall recoil pressure under passive conditions.

WOB

0.35-2.4 j min-1 [44]

PTP

80-200 cmH2O s min-1 [43]

Negative airway pressure swing during end expiratory occlusion

ΔPocc

6-13 cmH2O [46]

Good coorelation with ΔPes and Pmus, easy to obtain in intubated patients.

Requires intubation and collaboration of the patient.

Tidal swing of CVP

ΔCVP

0-8 mmHg [49]

Good correlation with ΔPes, useful in the absence of an esophageal balloon catheter

Depends on volemic state of the patient

Diaphragm ultrasound

Thickening fraction

15-30% [51]

Easy to obtain at the bedside and in non-intubated patients, cheap.

Does not account for inspiratory and expiratory muscles activation

Reviewer 2 Report

Comments and Suggestions for Authors

This is an interesting paper. Minor changes should be made.

1.      ABSTRACT: Explain what does it mean with ARDS. Acronyms should be explained the first time they are introduced.

2.      When introducing the term "occult pendelluft"  explain it so a person unfamiliar with it can understand it.

3.      Do the same with "vicious cycle of patient self-inflicted lung injury (P-SILI)

4.      In Table 1, introduce a column with the numeric value of the effect; for example, in reference 17, there is a reduction in mortality and an increase in respiratory-free days.

5.      As the paper presents complex physiological processes,  authors should include son graphs , flow charts, or drawings to facilitate understanding.

6.      In the introduction, write a sentence explaining the magnitude of the problem, for example, how many patients a year there are in ICU care in the world, Europe or Italy, how many are with invasive mechanical ventilation, etc., to attract the interest of the reader, and allow to have an insight of the magnitude of the problem.

Author Response

  1. ABSTRACT: Explain what does it mean with ARDS. Acronyms should be explained the first time they are introduced.

Done. We didn’t notice it, thank you.

  1. When introducing the term "occult pendelluft"  explain it so a person unfamiliar with it can understand it.

We now added the explanation of the term occult pendelluft.

Line 94: “Occult pendelluft” is the shift of gas from non-dependent to dependent regions during inspiration, in addition to the dorsal tidal volume coming from outside (ventilator or non-invasive support).

  1. Do the same with "vicious cycle of patient self-inflicted lung injury (P-SILI)

We now better explained this concept.

Line 66: “Vicious cycle” refers to P-SILI auto-maintenance: high respiratory drive induces high effort deter-mining high PL, which worsens lung injury. The consequences of worsening lung injury (inflammation, alterations in gas exchange, acidosis, cyclic derecruitment) further trigger respiratory drive and the cycle re-starts (Figure 1).

  1. In Table 1, introduce a column with the numeric value of the effect; for example, in reference 17, there is a reduction in mortality and an increase in respiratory-free days.

Thank you for the suggestion. The table already describes these data and provides the references for the reader to further explore the details.

  1. As the paper presents complex physiological processes,  authors should include son graphs , flow charts, or drawings to facilitate understanding.

We now added an exemplificative figure about P-SILI with the main mechanisms involved in its development. See figure 1.

Figure 1. The figure shows the “vicious cycle” of P-SILI. Lung injury triggers alterations in gas exchange, derecruitment and activation of inflammation, all contributing to an increase in the respiratory drive. High respiratory drive produces high effort which in turns causes large deflections in pleural pressure with lung stress and hemodynamic changes, resulting in volu- and barotrauma, atelectrauma, lung edema and vascular shear stress. All these mechanisms further increase lung injury, and the cycle re-starts. PL,transpulmonary pressure; RV, right ventricle; P-SILI, patient's self inflicted lung injury. [Created with BioRender.com].

  1. In the introduction, write a sentence explaining the magnitude of the problem, for example, how many patients a year there are in ICU care in the world, Europe or Italy, how many are with invasive mechanical ventilation, etc., to attract the interest of the reader, and allow to have an insight of the magnitude of the problem.

We now reported data from the LUNG SAFE, a multicenter prospective cohort study on epidemiology of ARDS to explain the magnitude of the problem.

Line 26: This is particularly relevant in patients with acute respiratory distress syndrome (ARDS), which accounts for ~ 10% of total admissions to intensive care and 23% of all mechanically ventilated patients [1]. According to the LUNG SAFE multicenter prospective cohort study, around 30% of patients with ARDS are spontaneously breathing already on the first day of admission to the ICU [1]. Fine tuning spontaneous breathing and, before, the timing of intubation may be crucial to the prognosis of these patients [2].